SCIENCE FORUM

# Donated chemical probes for open science

**Abstract** Potent, selective and broadly characterized small molecule modulators of protein function (chemical probes) are powerful research reagents. The pharmaceutical industry has generated many high-quality chemical probes and several of these have been made available to academia. However, probe-associated data and control compounds, such as inactive structurally related molecules and their associated data, are generally not accessible. The lack of data and guidance makes it difficult for researchers to decide which chemical tools to choose. Several pharmaceutical companies (AbbVie, Bayer, Boehringer Ingelheim, Janssen, MSD, Pfizer, and Takeda) have therefore entered into a pre-competitive collaboration to make available a large number of innovative high-quality probes, including all probe-associated data, control compounds and recommendations on use (https://openscienceprobes.sgc-frankfurt.de/). Here we describe the chemical tools and target-related knowledge that have been made available, and encourage others to join the project.

SUSANNE MÜLLER*, SUZANNE ACKLOO, CHERYL H ARROWSMITH, MARCUS BAUSER, JEREMY L BARYZA, JULIAN BLAGG, JARK BÖTTCHER, CHAS BOUNTRA, PETER J BROWN, MARK E BUNNAGE, ADRIAN J CARTER, DAVID DAMERELL, VOLKER DÖTSCH, DAVID H DREWRY, ALED M EDWARDS, JAMES EDWARDS, JON M ELKINS, CHRISTIAN FISCHER, STEPHEN V FRYE, ANDREAS GOLLNER, CHARLES E GRIMSHAW, ADRIAAN IJZERMAN, THOMAS HANKE, INGO V HARTUNG, STEVE HITCHCOCK, TREVOR HOWE, TERRY V HUGHES, STEFAN LAUFER, VOLKHART MJ LI, SPIROS LIRAS, BRIAN D MARSDEN, HISANORI MATSUI, JOHN MATHIAS, RONAN C O'HAGAN, DAFYDD R OWEN, VINEET PANDE, DANIEL RAUH, SAUL H ROSENBERG, BRYAN L ROTH, NATALIE S SCHNEIDER, CORA SCHOLTEN, KUMAR SINGH SAIKATENDU, ANTON SIMEONOV, MASAYUKI TAKIZAWA, CHRIS TSE, PAUL R THOMPSON, DANIEL K TREIBER, AMÉLIA YI VIANA, CARROW I WELLS, TIMOTHY M WILLSON, WILLIAM J ZUERCHER, STEFAN KNAPP AND ANKE MUELLER-FAHRNOW*

## "Man must shape his tools lest they shape him" (Arthur Miller)

The function of a protein can be explored in several different ways. Genetic approaches are used to suppress the expression of the respective gene/protein, for example using gene editing methods such as siRNA or shRNA or by CRISPR/Cas9 (*Mali et al., 2013*). However, in drug discovery, these methods have some deficiencies: they commonly remove or suppress the entire protein and thus cannot easily reveal the function of a specific druggable protein domain – although domain-based CRISPR is becoming a more widely used method; they are not reversible; their effects are not instantaneous; and they not only disrupt the protein, but also the protein interactome around the targeted protein. Selective small molecule modulators ('chemical probes'), in contrast, can probe the particular function of a targeted domain and can, therefore, be used to study its role in biological processes and in human disease in a dose and time-dependent manner across a wide range of cell and animal models. These probes can also be modified to enhance the degradation of the protein(s) they bind to (*Mali et al., 2013*; *Toure and Crews, 2016*).

Small molecules can be used in a broad panel of assay systems comprising primary cells, tissues and also in vivo models, and other systems not easily amenable even for state-of-the-art genetic target validation methods. Despite the fact that non-selective compounds cast a wide net and can be used to uncover interesting poly-pharmacologies, having a panel of selective probes that can be used in combination will facilitate data deconvolution and target identification. These properties, together with the possibility of further development of probes into drug candidates, make them among the most versatile tools to explore the relevance of a protein for therapeutic development. However, the necessary characterization data is often missing for chemical compounds, and inhibitors are announced as being 'selective' despite missing a comprehensive profile. Tool compounds, which are chemically unstable or not comprehensively characterized are therefore limited in their utility (*Arrowsmith et al., 2015*). Moreover, poorly characterized chemical modulators generate misleading results and litter the literature with contradicting data on a target's function and its role in biology. This is also true for probes that are used improperly, e.g. at higher than appropriate concentration thus inhibiting other proteins in addition to the target or resulting in non-specific cellular toxicity. Unfortunately,

reactive and non-specific inhibitors are widely used in the academic research community, often resulting in incorrect functional annotation (*Baell and Walters, 2014*).

The ideal chemical probes need to be selective, active in cells and chemically stable. The recent discussion on best practice within the chemical biology community suggested a number of stringent quality criteria for chemical probes (*Arrowsmith et al., 2015*; *Blagg and Workman, 2017*; *Edwards et al., 2009*; *Bunnage et al., 2013*). Typical criteria as applied by the Structural Genomics Consortium (SGC) are shown in *Figure 1*, although these may vary slightly depending on the specific protein.

A diverse set of chemical tool compounds is available to cell biologists. However, characterization data associated with these compounds are often either incomplete or buried in patents or supplemental data files of publications. Thus, scientists face a challenge to decide which tools to use for their research. Help is provided for example by the Chemical Probes Portal (*Baell and Walters, 2014*; *Blagg and Workman, 2017*), which was established in 2015 to provide a comprehensive overview of published and newly released tool compounds that are annotated with a simple star-rating system. All compounds submitted to the portal are reviewed by at least three members of an independent

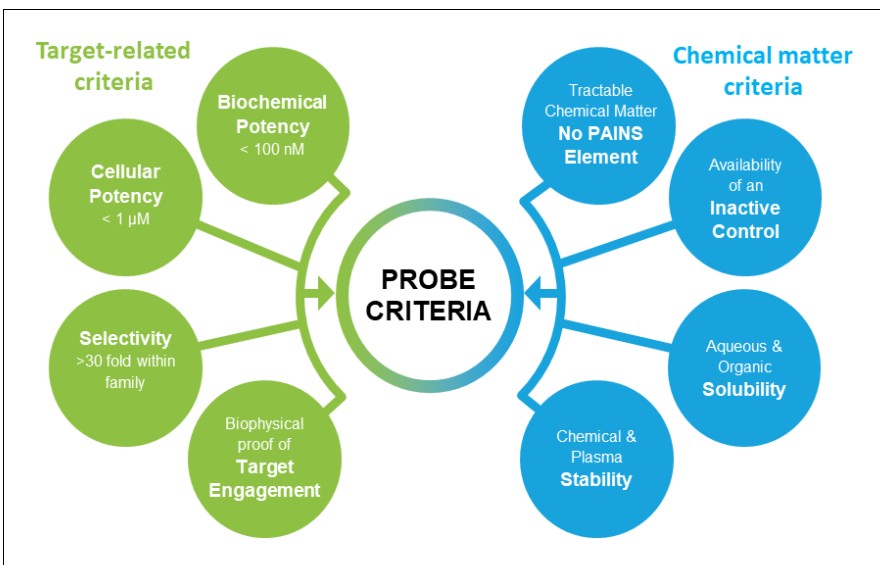

**Figure 1.** Chemical probes need to fulfil stringent criteria to qualify as research tools. Shown here are target and compound related criteria applied by the Structural Genomics Consortium.
DOI: https://doi.org/10.7554/eLife.34311.002

expert scientific advisory board. Only probes that receive three stars ('Best available probe for this target, or a high-quality probe that is a useful orthogonal tool') or four stars ('Recommended as a probe for this target') are recommended to be used. Of all the compounds submitted to the probe portal so far (about 400), 125 have achieved a rating of three stars or better, thereby showing that there is an urgent need for more high-quality tool compounds to foster reproducible research.

## "Excellence, then, is not an act, but a habit" (Will Durant [Durant, 1926])

Like drug discovery, probe development is a multi-disciplinary effort involving experts from several areas including protein chemistry, biochemistry, cell biology, pharmacology and medicinal chemistry (*Dahlin and Walters, 2014*; *Garbaccio and Parmee, 2016*). Once a target has been selected, the first step is the design of a project-specific screening cascade. The screening procedure needs to reflect target-related probe criteria as well as the desired compound properties.

A typical screening cascade for a kinase probe discovery project is shown in *Figure 2*. The screening cascade consists of a primary assay – usually a biochemical activity assay – plus an assay with an orthogonal readout, e.g., a

biophysical assay, a number of selectivity assays for the target and a cell-based assay to demonstrate on-target activity in the cellular environment. If possible, a crystallization system should be established to elucidate the binding modes of selected compounds enabling the rational design of better inhibitors. In silico analyses to exclude undesired events such as frequent hitters and pan-assay interference compounds (PAINs), and assays characterizing the physical and chemical properties of the identified hits (*Hughes et al., 2011*) complement the analysis (*Baell and Walters, 2014*). Medicinal chemistry optimization is then started for selected compound classes. For a typical project, multiple rounds of the Design – Make – Test – Analyze circle (*Plowright et al., 2012*) are needed before a suitable probe candidate is identified. Importantly, cross-correlating results from different assays within a compound class (e.g. tracking of cellular read-outs with biochemical potency and cellular target engagement data) provide a continuing consistency check if observed effects are truly a function of inhibiting the target of interest. Experience within the SGC shows that approximately 1–2 years and €2 million are needed to generate one chemical probe fulfilling these stringent criteria (*Donner, 2014*). This observation is in line with the experience of many medicinal chemists at pharmaceutical companies.

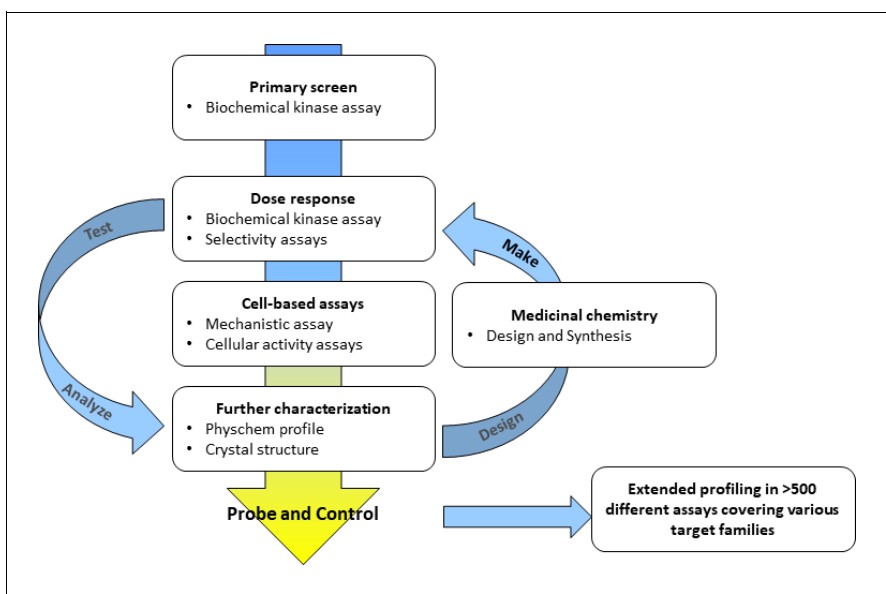

**Figure 2.** Typical workflow for a kinase probe discovery project. Medicinal chemistry optimization involving multiple iterative steps of compound design, synthesis and screening are necessary until probe criteria are fulfilled.

DOI: https://doi.org/10.7554/eLife.34311.003

As there may be similarities among binding sites on both related and unrelated proteins, unwanted binding to proteins other than the original target is regularly observed. This selectivity challenge can never be completely avoided, and thus the end users should be aware of unknown cross-reactivity challenges. A way to reduce the risk of non-specific effects is the use of a suitable control compound having a chemical structure closely related to that of the probe but lacking activity on the target. A wider profiling against a panel of pharmacologically active targets or proteomics analysis of the compound provides additional information about compound selectivity. Having access to multiple probes from structurally different chemical series further reduces the risk that unknown off-target activities give rise to incorrect conclusions about target functions.

## "Well begun is half done" (Aristotle)

Many quality probe compounds are buried in the chemical vaults of the pharmaceutical industry, depriving the scientific community of useful tools and limiting the impact of the original research. In some cases, particular compounds, their properties and some structure–activity relationships (SAR) have been published (*Nara et al., 2014*; *Siebeneicher et al., 2016*; *Takahashi et al., 2015*; *Wu-Wong et al., 1999*). However, often only selected data are published and the proprietary compounds are not made available to the researchers except via restrictive contractual agreements, and this impedes their use and their impact. Indeed, in the nuclear hormone receptor field, we showed that any legal encumbrances to compound access reduced the subsequent use of the compound in the literature significantly (*Isserlin, 2011*). Thus, the open access/open science approach is the fastest route to reach the end users and thereby to have a positive effect on research.

This evidence, as well as impact from the SGC epigenetics probes project, has convinced the SGC partner companies that the release of previously hidden compounds and data to the public will provide value to science and to the companies (*Lee, 2015*). To this end, seven pharmaceutical companies associated with the SGC have each agreed to donate 10 of these valuable compounds, stemming from their research pipelines, for a total of 70 high-quality small molecules, thus providing a major boost to the chemical biology toolbox. The compounds have

been selected based on a variety of criteria, which are different for each participating company. These include profiling available for the compound, feasibility of generating a control compound, availability of physical compound, target class, intellectual property considerations, and other factors.

This is an exciting development, but many of the compounds will require wider profiling to meet today's more stringent quality criteria. As the primary focus of the pharmaceutical industry is not to generate chemical probes, but to develop new drugs, not all donated probes have been profiled to the same depth that is required of a high-quality chemical probe. Moreover, specificity for a particular target is not a requirement for an effective drug. Thus, although most of the pharma-donated probes have been extensively characterized, they often need to be better adapted for use as a single chemical tool (*Figure 1*). In particular, no bespoke control compounds have been generated as the progress of the probe compounds within the company is usually followed by extensive SAR across a series of analogues. Selection and characterization of the control compound is needed to complete a probe package. In addition, control compounds also have to be carefully characterized to weed out promiscuous compounds.

The aim of our partnership is to provide this comprehensive characterization. We believe this to be a valuable contribution to the community. Once broadly characterized and accompanied by relevant control compounds, the initial set of 70 probes reflect a collective contribution of at least €140 million to the public domain (*Figure 3*). These donated probes cover a broad array of targets from different protein families relevant for a number of disease indications (see *Table 1*).

In order to guarantee the quality of the compounds, the donated probe candidates and control compounds are subjected to a two-tier scientific review process: the first review takes place internally, including partners who have not been involved in the probe project, and the second review is performed by a panel of renowned scientists, who have agreed to act as independent reviewers. The first 30 proposals were presented to the internal review committee during a two-day meeting in June 2017 in Frankfurt am Main, Germany, where a process for their release to the public was also established. At this 'historic' meeting scientists from eight pharmaceutical companies scrutinized the quality of the probes proposed by the other partners and

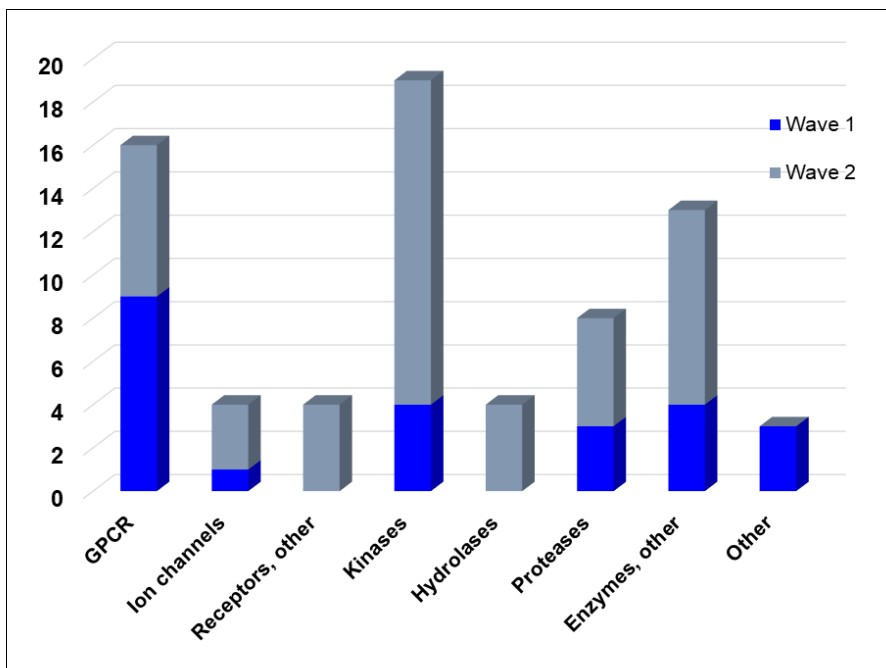

**Figure 3.** Overview of targets for which pharmaceutical companies have volunteered to donate chemical probes. Planned release for wave one probes is in spring 2018 pending the outcome of independent peer review. The targets of this first wave of probes are given in **Table 1**. Final numbers may slightly vary as some chemical probes are still in the approval process.
DOI: https://doi.org/10.7554/eLife.34311.004

made constructive suggestions on improvement of the associated data packages (*Figure 4A*).

In the initial set, most targets are uniquely addressed by only one chemical compound, but a maximum of two chemical probes for the same target will be accepted if they represent different chemotypes as judged by the review panels. The remaining probe sets will be provided during the course of 2018/2019. All approved probes are measured against the same quality criteria (*Figure 1*) and will be profiled in assay panels comprising of >500 assays, including broad panels of pharmacologically active targets such as GPCRs, kinases, ion channels and proteases to identify off-target activities (*Table 2*). Disease-specific phenotypic panels such as assays in primary tissues established by SGC partners will provide an initial characterization of their biological effects (*Edwards et al., 2015*).

The proposed probes range from completely novel 'best in class', to probes that have been selected because they are provided as a complete set, with control compounds. Although some of the proposed compounds themselves are already commercially available, for most there is no widely characterized partner control compound (*Figure 4B*).

The current probe proposals cover proteins from many different families such as GPCRs, kinases and proteases as well as other protein targets implicated in a variety of therapeutic areas ranging from oncology to inflammatory diseases and neurodegenerative disorders. An excellent example of a donated probe is the recently published p300/CBP histone acetyltransferase (HAT) inhibitor (A-485), which was shown to have efficacy in several cell models of malignancies (*Lasko et al., 2017*). This probe, including its control compound, has been approved by both internal as well as external reviewers and is now available to the scientific community. In contrast, other donated probes are not published or only mentioned in patents and therefore have not been accessible at all. Examples include a novel coagulation factor II thrombin receptor (F2R/PAR-1) inhibitor, which has potential for thrombosis management, and an inhibitor for focal adhesion kinase (FAK) and proline-rich tyrosine kinase 2 (PYK2), which has been in clinical trials for advanced non-haematologic malignancies, but for which profiling data have not yet been available. Even previously published probes are not always widely accessible. For example, the set includes a probe for the solute carrier NHE1, a target associated with ischemia/reperfusion-

**Table 1.** Targets of first wave of donated probes (approved or close to approval).

| Family | Target | Mode of action | Company | Structure |
|---|---|---|---|---|
| | D$_4$ Dopamine receptor | Agonist | AbbVie ABT-724 | |
| GPCR | ET$_A$ Endothelin receptor | Antagonist | AbbVie ABT-546 | |
| | Par1/F2R (F2R) Protease activated receptor | Antagonist | Bayer BAY-386 | |
| | CRTH2 (Prostaglandin DP$_2$ receptor) | Antagonist | MSD CRTH2i | |
| | CB$_1$ Cannabinoid receptor | Inverse Agonist | MSD MRL-650 | |
| | EP$_2$Prostaglandin receptor | Antagonist | Pfizer PF-04418948 | |
| | α$_{1D}$ Adrenoceptor | Antagonist | Takeda (R)-9s | |
| | KISS1 Receptor (GPR54) | Agonist | Takeda KISS1-305 | D-Tyr-D-Pya(4)-Asn-Ser-Phe-azaGly-Leu-Arg(Me)-Phe-NH2 |

*Table 1 continued on next page*

*Table 1 continued*

| Family | Target | Mode of action | Company | Structure |
|---|---|---|---|---|
| Hydrolase | sEH (Soluble epoxide hydrolase) | Inhibitor | Boehringer Ingelheim BI-1935 | |
| | FAAH (Fatty acid amide hydrolase) | Inhibitor | Pfizer PF-04457845 | |
| Ion channel | TRPM8 (Cold and menthol receptor 1) | Antagonist | Pfizer PF-05105679 | |

*Table 1 continued on next page*

*Table 1 continued*

| Family | Target | Mode of action | Company | Structure |
|---|---|---|---|---|
| Kinase | c-MET (Tyrosine-protein Kinase Met) | Inhibitor | Bayer BAY-474 | |
| | TIE (Tyrosine kinase with Ig and EGF homology domains 1), DDR (Discoidin domain receptor family) | Inhibitor | Bayer BAY-826 | |
| | ERK1/2 (Extracellular signal-regulated kinase) | Inhibitor | MSD MRK-ERKi | |
| | SYK (Spleen tyrosine kinase) | Inhibitor | MSD MRL-SYKi | |
| | FAK/PYK2 (focal adhesion kinase /proline-rich tyrosine kinase 2) | Inhibitor | Pfizer PF-04554878 | |

*Table 1 continued on next page*

*Table 1 continued*

| Family | Target | Mode of action | Company | Structure |
|--------|--------|----------------|---------|-----------|
| | FLAP (5-Lipoxygenase-activating protein) | Inhibitor | Boehringer Ingelheim BI 665915 | |
| Other | FASN (Fatty acid synthase) | Inhibitor | Boehringer Ingelheim BI 99179 | |
| | MIF (Macrophage migration inhibitory factor) | Activator | Takeda BTZO-1 | |
| | Farnesyltransferase | Inhibitor | AbbVie ABT-100 | |
| | P300/CBP (E1A binding protein/ CREB binding protein) | Inhibitor | AbbVie A-485 | |
| | NHE1, SLC9A1 | Antagonist | Boehringer Ingelheim BI-9267 | |
| | MTH1 (MutT homolog 1) | Inhibitor | Bayer BAY-707 | |

*Table 1 continued on next page*

*Table 1 continued*

| Family | Target | Mode of action | Company | Structure |
|---|---|---|---|---|
| Protease | MMP12 (Matrix metallopeptidase 12) | Inhibitor | Bayer BAY-7598 | |
| | Gamma secretase | Inhibitor | MSD MRK-560 | |
| | Gamma secretase | Modulator | MSD GSM1 | |
| | METAP2 (Methionine aminopeptidase-2) | Inhibitor | Takeda TP-004 | |

DOI: https://doi.org/10.7554/eLife.34311.005

induced cell death, a peptidomimetic agonist for the KISS1 receptor, which plays a crucial role in cellular hormone function and puberty, and the inhibitor for a gamma secretase (GSI) protease, which may have potential in targeting Alzheimer's disease.

Using the infrastructure and established processes of past SGC probe projects, a non-bureaucratic and simple distribution process is implemented. This process involves distribution in bespoke probe libraries under a simple web-accessible Open Science Trust Agreement (http://www.thesgc.org/click-trust) as well as through trusted commercial vendors. To the best of our knowledge, our initiative is unique in enabling open access to well-validated probes including controls generated in the pharmaceutical industry for diverse target families.

All supporting potency and selectivity data, as well as advice for the appropriate use of the compounds for cellular assays and – if applicable – in vivo assays, will be easily accessible via the public database (https://openscienceprobes.sgc-frankfurt.de/). The launch for the first version is planned for the beginning of 2018. The database supports the data needs of both biologists and chemists. The first version focusses on a search for the target proteins, probes, control compounds and recommendations on use. For the second version, additional features such as chemical substructure searches will be accessible. Full assay details will be provided and reagents used will be listed so that scientists using the probes are enabled to judge the quality of the data provided as well as to reproduce key data in their own lab. For example, it is important to know if a protein kinase has been screened in a binding or activity assay, and which ATP concentration has been used. Further, the protein construct used to perform certain assays is of significance.

As both the probes and the negative controls will be characterized in more than 500 assays, we will generate more than 70,000 biological data sets within the next 1–2 years: a rich and easily accessible source for future analyses. By

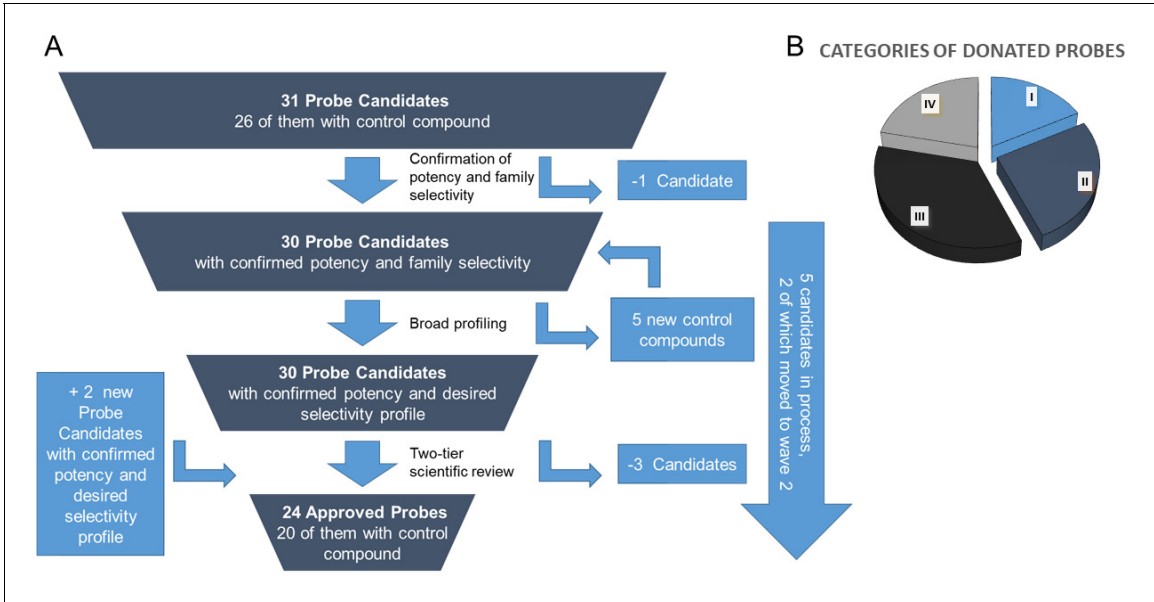

**Figure 4.** Attrition rate and categories of donated probes.

DOI: https://doi.org/10.7554/eLife.34311.006

A. Attrition rate of the approval process of the proposed chemical probes. B. Approved probes were categorized to show their differentiation from available chemical modulators (i) targets for which there are currently no high-quality probes available; (ii) targets for which the donated probe promises a significant (e.g. 10-fold) benefit in potency or selectivity; (iii) cases in which the new donated probe has similar potency/selectivity as currently available probes but an entirely different chemotype; (iv) best in class compound where none of the above points apply and where the benefit lies in the availability of the control compound and/or the data annotation.

providing the data in a comprehensive way we hope to extend our understanding of this particular mechanism or protein in a way that leads to new therapeutic approaches.

The new tool compounds and the corresponding data will help to improve the quality of research and will deepen our understanding of the target biology. However, comprehensive characterization, which ideally should be consistent to make data comparable and facilitate data mining, comes at a cost, and in many cases also requires resources for the (re-)synthesis of the chemical probe. The biggest problem is in the availability and characterization of the

**Table 2.** Overview of data generated for all donated probes.

| Assays | Scope | Timing |
|---|---|---|
| Target-specific assays (biochemical/ biophysical/ cell-based) | All chemical probes & controls | Before release (decision criteria) |
| Target-specific selectivity panels | | |
| 500+ kinases | All chemical probes & controls | After release (annotation) |
| Broad specificity panel, 100+ ion channels, GPCRs, proteases | | |
| 30+ epigenetics targets | | |
| Phenotypic assays (cell lines & primary human material) | | |
| 3D structure of protein-ligand complex | Subset | Optional |
| Physchem parameters, e.g. solubility | Subset | |
| *In vivo* experiments | Selected probes | |

These data will be made available through a publicly available database.

DOI: https://doi.org/10.7554/eLife.34311.007

control compounds. Not all candidate probes will have a suitable similar analogue that does not inhibit/activate the target of interest. Funding will be necessary in order to fill such gaps, but financing such a project and the resources to synthesize and characterize compounds is no easy task. It does not usually fit into the remit of the major funding agencies and help is needed by relevant translational organizations and individual labs to support the project. Organizations like the Division of Pre-Clinical Innovation of the National Center for Advancing Translational Sciences (NCATS; https://ncats.nih.gov/) and the National Institute of Mental Health (NIMH) Psychoactive Drug Screening Program (https://pdspdb.unc.edu/pdspWeb/) housed at the laboratory of Dr Bryan Roth at UNC are generously supporting this crucial endeavour by synthesizing negative controls and conducting probe profiling experiments.

The distribution of probes will occur via commercial vendors, but it is no trivial task to make the well-characterized control compounds available to the user. Due to reduced revenues from the control compounds, vendors are often reluctant to offer these important controls. Regrettably, researchers often perform experiments without the appropriate control compound due to cost reasons or because initial experiments have already been performed without the control. A trial kit, which we will offer, including both probe and control compound, and/or sets of pre-diluted compounds may aid researchers to perform properly controlled experiments from the start. It is up to the combined efforts of researchers, vendors, journal editors and referees to make use of the chemical probes in combination with their available control standard practice in biomedical research.

## "From a small seed a mighty trunk may grow" (Aeschylus)

While in the past almost all aspects of pharmaceutical research and development (R&D) were seen as competitive, the thinking in the field has shifted remarkably over the last decade. More and more challenges in the R&D process are seen as precompetitive, resulting in public–private partnerships and multilateral, critical mass consortia jointly addressing overarching issues. Many pharmaceutical companies have initiated open innovation projects interacting with the academic community. A key success factor for these endeavours is the easy access to know-how and reagents without complicated

contractual arrangements (*Nilsson and Felding, 2015*; *Ehrismann and Patel, 2015*). We hope that the project initiated here will entice other companies and academics to follow suit and join us in the quest to increase the availability of well-validated probes meeting stringent quality criteria for the scientific community and decide to make some of their assets openly available. Whilst ultimately, the success of the project will depend on the willingness and support of the scientific community, additional pharmaceutical companies and funding bodies to engage, we believe this is an exciting first step in uncovering and delivering high–quality chemical probes to unlock new biology and ultimately new high-quality targets for drug discovery.

## Acknowledgements
The SGC is a registered charity (number 1097737) that receives funds from AbbVie, Bayer AG, Boehringer Ingelheim, Canada Foundation for Innovation, Eshelman Institute for Innovation, Genome Canada, Innovative Medicines Initiative (EU/EFPIA) [ULTRA-DD grant no. 115766], Janssen, Merck KGaA Darmstadt Germany, MSD, Novartis Pharma AG, Ontario Ministry of Economic Development and Innovation, Pfizer, São Paulo Research Foundation-FAPESP, Takeda, and Wellcome [106169/ZZ14/Z]. This work was also partially funded by the DFG Cluster of Excellence for Macromolecular Complexes.

**Susanne Müller** is in the Structural Genomics Consortium, Buchmann Institute for Molecular Life Sciences, Goethe University Frankfurt, Frankfurt am Main, Germany
susanne.mueller-knapp@bmls.de
http://orcid.org/0000-0003-2402-4157

**Suzanne Ackloo** is in the Structural Genomics Consortium, University of Toronto, Toronto, Ontario, Canada

**Cheryl H Arrowsmith** is in the Structural Genomics Consortium and the Princess Margret Cancer Centre, Department of Medical Biophysics, University of Toronto, Toronto, Ontario, Canada

**Marcus Bauser** is at Bayer AG, Drug Discovery Pharmaceuticals, Berlin, Germany

**Jeremy L Baryza** is at Vertex Pharmaceuticals, Boston, Massachusetts, United States

**Julian Blagg** is in the Cancer Research UK Cancer Therapeutics Unit, The Institute of Cancer Research, London, United Kingdom

**Jark Böttcher** is at Boehringer Ingelheim, Discovery Research, Vienna, Austria

**Chas Bountra** is in the Structural Genomics Consortium, Nuffield Departmentof Medicine, University of Oxford, Oxford, United Kingdom

**Peter J Brown** is in the Structural Genomics Consortium, University of Toronto, Toronto, Ontario, Canada

**Mark E Bunnage** is at Vertex Pharmaceuticals, Boston, Massachusetts, United States

**Adrian J Carter** is at Boehringer Ingelheim, Discovery Research, Ingelheim am Rhein, Germany

**David Damerell** is in the Structural Genomics Consortium, Nuffield Department of Medicine, University of Oxford, Oxford, United Kingdom

**Volker Dötsch** is a Reviewing Editor of eLife and is in the Institute of Biophysical Chemistry and Center for Biomolecular Magnetic Resonance, Goethe University, Frankfurt am Main, Germany
https://orcid.org/0000-0001-5720-212X

**David H Drewry** is in the Structural Genomics Consortium, UNC Eshelman School of Pharmacy, University of North Carolina at Chapel Hill, Chapel Hill, North Carolina, United States

**Aled M Edwards** is in the Structural Genomics Consortium, University of Toronto, Toronto, Ontario, Canada

**James Edwards** is at Janssen Pharmaceutical Research and Development LLC, Spring House, Pennsylvania, United States

**Jon M Elkins** is in the Structural Genomics Consortium, Nuffield Department of Medicine, University of Oxford, Oxford, United Kingdom and the Structural Genomics Consortium, Universidade Estadual de Campinas — UNICAMP, Campinas, Brazil
https://orcid.org/0000-0003-2858-8929

**Christian Fischer** is at Merck & Co., Inc., Boston, Massachusetts, United States

**Stephen V Frye** is in the Center for Integrative Chemical Biology and Drug Discovery, Division of Chemical Biology and Medicinal Chemistry, UNC Eshelman School of Pharmacy, University of North Carolina at Chapel Hill, Chapel Hill, North Carolina, United States

**Andreas Gollner** is at Boehringer Ingelheim, Discovery Research, Biberach an der Riss, Germany

**Charles E Grimshaw** is a Pharma and Biotech Consultant at Ched Grimshaw Consulting, LLC, Poway, San Diego, California, United States
https://orcid.org/0000-0002-2897-3483

**Adriaan IJzerman** is in the Division of Medicinal Chemistry, LACDR, Leiden University, Leiden, The Netherlands

**Thomas Hanke** is in the Structural Genomics Consortium, Buchmann Institute for Molecular Life Sciences, Goethe University Frankfurt, Frankfurt am Main, Germany
https://orcid.org/0000-0001-7202-9468

**Ingo V Hartung** is at Bayer AG, Drug Discovery Pharmaceuticals, Berlin, Germany

**Steve Hitchcock** is at Takeda California Inc., San Diego, California, United States

**Trevor Howe** is at J&J Innovation Centre, London, United Kingdom

**Terry V Hughes** is at J&J Innovation Centre, London, United Kingdom

**Stefan Laufer** is in the Department of Pharmaceutical Chemistry, Eberhard Karls Universität Tübingen, Tübingen, Germany

**Volkhart MJ Li** is at Bayer AG, Drug Discovery Pharmaceuticals, Wuppertal, Germany

**Spiros Liras** is at Worldwide Medicinal Chemistry, Pfizer, Cambridge, Massachusetts, United States

**Brian D Marsden** is in the Structural Genomics Consortium, Nuffield Department of Medicine, and the Kennedy Institute of Rheumatology, Nuffield Department of Orthopaedics, Rheumatology and Musculoskeletal Sciences University of Oxford, Oxford, United Kingdom

**Hisanori Matsui** is at Research Takeda Pharmaceutical Company Ltd., Fujisawa, Japan

**John Mathias** is at Worldwide Medicinal Chemistry, Pfizer, Cambridge, Massachusetts, United States

**Ronan C O'Hagan** is at Merck & Co., Inc., Boston, Massachusets, United States

**Dafydd R Owen** is at Worldwide Medicinal Chemistry, Pfizer, Cambridge, Massachusets, United States

**Vineet Pande** is at Discovery Sciences, Janssen-Pharmaceutical Companies of Johnson & Johnson, Beerse, Belgium

**Daniel Rauh** is in the Fakultät für Chemie und Chemische Biologie, Technische Universität Dortmund, Dortmund, Germany

**Saul H Rosenberg** is at AbbVie, North Chicago, Illinois, United States

**Bryan L Roth** is in The National Institute of Mental Health Psychoactive Active Drug Screening Program and the Department of Pharmacology, University of North Carolina Chapel Hill School of Medicine, Chapel Hill, North Carolina, United States

**Natalie S Schneider** is in the Structural Genomics Consortium, Buchmann Institute for Molecular Life Sciences, Goethe University Frankfurt, Frankfurt am Main, Germany

**Cora Scholten** is at Bayer AG, Drug Discovery Pharmaceuticals, Berlin, Germany

**Kumar Singh Saikatendu** is at Takeda California Inc., San Diego, California, United States

**Anton Simeonov** is in the National Center for Advancing Translational Sciences, National Institutes of Health, Bethesda, Maryland, United States

**Masayuki Takizawa** is at Research Takeda Pharmaceutical Company Ltd., Fujisawa, Kanagawa, Japan

**Chris Tse** is at AbbVie, North Chicago, Illinois, United States

**Paul R Thompson** is in the Department of Biochemistry and Pharmacology and the Program in Chemical Biology, University of Massachusetts Medical School, Worcester, United States

**Daniel K Treiber** is at Eurofins DiscoverX, San Diego, California, United States

**Amélia YI Viana** is at Boehringer Ingelheim, Ingelheim am Rhein, Germany

**Carrow I Wells** is in the Structural Genomics Consortium, UNC Eshelman School of Pharmacy, University of North Carolina at Chapel Hill, Chapel Hill, North Carolina, United States

**Timothy M Willson** is in the Structural Genomics Consortium, UNC Eshelman School of Pharmacy, University of North Carolina at Chapel Hill, Chapel Hill, North Carolina, United States

**William J Zuercher** is in the Structural Genomics Consortium, UNC Eshelman School of Pharmacy, University of North Carolina at Chapel Hill, Chapel Hill, North Carolina, United States

**Stefan Knapp** is in the Structural Genomics Consortium, Buchmann Institute for Molecular Life Sciences, Goethe University Frankfurt, Frankfurt am Main, Germany

**Anke Mueller-Fahrnow** is at Bayer AG, Berlin, Germany

anke.mueller-fahrnow@bayer.com

*Competing interests:* Volker Dötsch: Reviewing editor, *eLife*. Marcus Bauser, Ingo V Hartung, Volkhart MJ Li, Cora Scholten, Anke Mueller-Fahrnow: employee of Bayer AG. Jeremy L Baryza, Mark E Bunnage: employee of Vertex Pharmaceuticals. Jark Böttcher, Adrian J Carter, Andreas Gollner, Amélia YI Viana: employee of Boehringer Ingelheim. James Edwards: employee of Janssen Pharmaceutical Research and Development LLC. Christian Fischer, Ronan C O'Hagan: employee of Merck & Co., Inc. Charles E Grimshaw: employee of Ched Grimshaw Consulting, LLC. Steve Hitchcock, Kumar Singh Saikatendu: employee of Takeda California Inc. Trevor Howe, Terry V Hughes: employee of J&J Innovation Centre. Spiros Liras, John Mathias, Dafydd R Owen: employee of Pfizer. Hisanori Matsui, Masayuki Takizawa: employee of Takeda Pharmaceutical Company Ltd. Vineet Pande: employee of Janssen-Pharmaceutical Companies of Johnson & Johnson. Daniel Rauh: Daniel Rauh received consulting and lecture fees (Sanofi-Aventis, Takeda, Novartis, Pfizer, LDC) as well as research support (Novartis, J&J, Bayer, Merck, MSD). Saul H Rosenberg, Chris Tse: employee of AbbVie. Daniel K Treiber: employee of Eurofins DiscoverX. The other authors declare that no competing interests exist.

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
