## [Decision Letter]

Thank you for submitting your article "Donated Chemical Probes For Open Science" to *eLife* for consideration as a Feature Article. Your article has been reviewed by two peer reviewers, and the evaluation has been overseen by Emma Pewsey (as Associate Features Editor for *eLife*) and Peter Rodgers (the *eLife* Features Editor) as the Senior Editor. The reviewers have opted to remain anonymous.

The reviewers have discussed the reviews with one another and I have drafted this decision to help you prepare a revised submission.

Summary:

The manuscript describes an important initiative that has the potential to significantly improve the quality of pharmacological agents available and further strengthen the use of chemical probes in many areas of biology. Importantly, probe-associated data generated internally, best use practices and inactive control compounds will also be shared. Thus, it will be of wide interest to the readers of *eLife*.

Essential revisions:

1) Please describe clearly how novel the donated probes are. No structures or names are provided for the probes listed in Table 1. Therefore, it is unclear in what way the current initiative is merely a repurposing/rebranding of other similar efforts or provides truly novel compounds. For example, the AbbVie P300/CBP inhibitor appears to be compound A-485, published in Nature 2017 and is currently available as an SGC probe and from several vendors. What constitutes the donation in that case, given that the compound is available, and data are already published? Similarly, are the Pfizer compounds those already currently available from Sigma for the same targets? Are the Boehringer Ingelheim compounds a subset of the ones made available in the opnMe initiative? Or are all these novel additional compounds?

2) High-quality probes already exist for several of the targets listed. Please include an analysis of the superiority of the new donated probes, indicating:

i) Targets for which there are currently no high-quality probes available;

ii) Targets for which the donated probe promises a significant (e.g. 10-fold) benefit in potency or selectivity;

iii) Cases in which the new donated probe has similar potency/selectivity as currently available probes but an entirely different chemotype;

iv) Cases where none of the above points apply (but there might still be a benefit in availability/control compounds/data annotation).

3) The purpose of the manuscript needs to be defined more clearly. Currently it is not clear whether the manuscript is primarily about defining chemical probes or about bringing awareness to the new resource.

4) Please provide more discussion about how the probes have been selected. It also needs to be made clear that the standards used for selection are not a definition of what a chemical probe is. Although having potent and selective inhibitor probes is ideal, non-selective chemical probes can also provide powerful information and should also be mentioned as good chemical probes.

---

## [Author Response]

Essential revisions:1) Please describe clearly how novel the donated probes are. No structures or names are provided for the probes listed in Table 1. Therefore, it is unclear in what way the current initiative is merely a repurposing/rebranding of other similar efforts or provides truly novel compounds. For example, the AbbVie P300/CBP inhibitor appears to be compound A-485, published in Nature 2017 and is currently available as an SGC probe and from several vendors. What constitutes the donation in that case, given that the compound is available, and data are already published? Similarly, are the Pfizer compounds those already currently available from Sigma for the same targets? Are the Boehringer Ingelheim compounds a subset of the ones made available in the opnMe initiative? Or are all these novel additional compounds?

The donated compounds of this initial wave are a mixture of novel, already published, and commercially available compounds. However, even though some of the compounds themselves might be available from other sources, for none is available a comprehensive and standard dataset comprising potency, selectivity, and often pharmacology, and for none is available an inactive negative control. Our collecting and organizing these underlying data is what differentiates a “compound that is commercially available” and a donated probe. For all compounds, the distinction compared to other initiatives will be that they are peer-reviewed by a 2-Tier process by an internal committee comprising all participating industry partners and independent external referees to be suitable as probes i.e. fulfilling selectivity and specificity criteria. This quality bar is a distinct feature of the donated probe set. Indeed, during the course of the review process, some probes were found no longer to be the ‘best in class’ compounds and the external referees rejected these compounds as part of the probe set.

Our selection of these specific donated probes was deliberate. Characterization data for these compounds were already fairly complete, which enabled a set of about 30 compounds to be proposed in a relatively short time-frame. An excellent example is A-485, which has a complete dataset and a negative control compound, something that is missing from most compounds provided by vendors. A-485 has been donated to the community and the associated data will be made more easily searchable in the database. For example, currently the selectivity data are buried in the supplements of the Nature publication, which describes this probe. For other compounds, like the ones from Pfizer, no negative control compound had been available before. Full profiling data (such as kinome wide selectivity screens) will also be made available through this initiative.

To facilitate uptake and proper use by the biology community, each probe characterization package also contains clear instructions on use (e.g. suggested concentration ranges) and a matching control compound accompanies each probe molecule. The Boehringer Ingelheim compounds are also available via the complementary opnMe initiative. However, the opnMe platform also contains compounds that have not been through the mentioned 2-Tier approval process, which is a quality factor missing for compounds that are available, but that have not been evaluated against quality criteria.

We have introduced Figure 4A to explain the process better and added the probe compound structures to Table 1.

We believe therefore that by this process the best available chemical probes will be made available for each target.

2) High-quality probes already exist for several of the targets listed. Please include an analysis of the superiority of the new donated probes, indicating:i) Targets for which there are currently no high-quality probes available;ii) Targets for which the donated probe promises a significant (e.g. 10-fold) benefit in potency or selectivity;iii) Cases in which the new donated probe has similar potency/selectivity as currently available probes but an entirely different chemotype;iv) Cases where none of the above points apply (but there might still be a benefit in availability/control compounds/data annotation).

We thank the referee for this suggestion. An analysis grouping the probes into the suggested categories has been added as Figure 4B.

3) The purpose of the manuscript needs to be defined more clearly. Currently it is not clear whether the manuscript is primarily about defining chemical probes or about bringing awareness to the new resource.

While the main point of the manuscript is to make researchers – notably our bioscience colleagues, who are the most common probe user community – aware of the new resource, we would like at the same time to point out why the resource is necessary and useful: despite the efforts of other initiatives the basic problem remains that chemical probes are difficult to find among the large number of chemical modulators that are currently available, which are often less specific or poorly characterized. In order to achieve easier access to the most suitable chemical probes and their negative control compounds, we introduce a definition of quality criteria for each target family. We also would like to highlight that the project involves more than just making some of the already available compounds accessible and adding new ones. The characterization of each probe against homogenous sets of screening assays within and outside the target family, synthesis of control compounds, the two-step approval process and additional phenotypic data contributed from probe users in an open database is a considerable extension to the current policy of making published compounds available by chemical vendors.

4) Please provide more discussion about how the probes have been selected. It also needs to be made clear that the standards used for selection are not a definition of what a chemical probe is. Although having potent and selective inhibitor probes is ideal, non-selective chemical probes can also provide powerful information and should also be mentioned as good chemical probes.

The probes have been selected based on a variety of factors: characterization level of the compounds, availability of material, novelty, availability of control compound. We have added a discussion of the selection criteria to the manuscript.

We agree with the reviewer that well characterized non-selective chemical compounds can unravel interesting pharmacologies and be useful tool compounds. However, we would like to reserve the term ‘chemical probe’ for potent and selective, cell active chemical modulators and distinguish them from non-selective tool compounds/modulators. The probe project, in the definition of the manuscript, aims to make available compounds that are optimized to connect a phenotype to one or very few target(s) the compound modulates. It should not be mistaken with a compound that has been optimized for a phenotypic effect in a given assay. We have amended the manuscript using the term ‘probes’ consistently only for selective chemical modulators to reflect this better. Additionally, wording has been added to reflect that non-specific chemical modulators can also be valuable.